# Mediterranean, DASH, and Alternate Healthy Eating Index Dietary Patterns and Risk of Death in the Physicians’ Health Study

**DOI:** 10.3390/nu13061893

**Published:** 2021-05-31

**Authors:** Yash R. Patel, Jeremy M. Robbins, J. Michael Gaziano, Luc Djoussé

**Affiliations:** 1Lifespan Cardiovascular Institute, Warren Alpert Medical School at Brown University, Providence, RI 02903, USA; 2Department of Medicine, Division of Aging, Brigham and Women’s Hospital and Harvard Medical School, Boston, MA 02120, USA; jeremy.robbins1@gmail.com (J.M.R.); jmgaziano@partners.org (J.M.G.); ldjousse@rics.bwh.harvard.edu (L.D.); 3Massachusetts Veterans Epidemiology and Research Information Center (MAVERIC) and Geriatric Research, Education and Clinical Research Center (GRECC), Boston Veterans Affairs Healthcare System, Boston, MA 02120, USA

**Keywords:** diet patterns, mortality, alternate healthy eating index, Mediterranean, DASH

## Abstract

Objective: Our primary objective was to examine the associations of the Mediterranean (MED), the Dietary Approaches to Stop Hypertension (DASH), and the Alternate Healthy Eating Index (AHEI) diet with total mortality. Our secondary objective was to examine the association of these three dietary patterns with cardiovascular disease (CVD) and cancer mortality. Research: Design and Methods: We prospectively studied 15,768 men from the Physicians’ Health Study who completed a semi-quantitative food-frequency questionnaire. Scores from each dietary pattern were divided into quintiles. Multivariable Cox regression models were used to estimate hazard ratio’s (95% confidence intervals) of mortality. Results: At baseline, average age was 65.9 ± 8.9 years. There were 1763 deaths, including 488 CVD deaths and 589 cancer deaths. All diet scores were inversely associated with risk for all-cause mortality: Hazard ratios (95% CI) of all-cause mortality from lowest to highest quintile for MED diet were 1.0 (reference), 0.85 (0.73–0.98), 0.80 (0.69–0.93), 0.77 (0.66–0.90), and 0.68 (0.58–0.79); corresponding values were 1.0 (reference), 0.96 (0.82–1.12), 0.95 (0.82–1.11), 0.88 (0.75–1.04), and 0.83 (0.71–0.99) for DASH diet and 1.0 (reference), 0.88 (0.77–1.02), 0.82 (0.71–0.95), 0.69 (0.59, 0.81), and 0.56 (0.47–0.67) for AHEI diet, after adjusting for age, energy, smoking, exercise, BMI, hypertension, coronary heart disease, congestive heart failure, diabetes, and atrial fibrillation. For cause-specific mortality, MED and AHEI scores were inversely associated with lower risk for CVD mortality, whereas AHEI and MED scores were inversely associated with lower risk for cancer mortality. Conclusion: Within this cohort of male physicians, AHEI, MED, and DASH scores were each inversely associated with mortality from all causes.

## 1. Introduction

Recognized healthy dietary patterns include the Mediterranean (MED) diet and the Dietary Approach to Stop Hypertension (DASH) diet, and these are core elements of the Alternative Healthy Eating Index (AHEI), a modified version of the United States Department of Agriculture’s healthy eating diet score [1,2].

Reedy et al. compared the relationship between the Healthy Eating Index-2010 (HEI-2010), the Alternative Healthy Eating Index-2010 (AHEI-2010), the Alternative Mediterranean Diet (aMED), and DASH and all-cause, cardiovascular disease (CVD) and cancer mortality in the National Institute of Health-AARP (NIH-AARP) Diet and Health Study, and found that higher scores for each dietary pattern were inversely associated with CVD and cancer mortality [3]. In the Multiethnic Cohort (MEC) study, Harmon et al. found that high HEI-2010, AHEI, aMED, and DASH scores were all inversely associated with risk of mortality from all causes, CVD, and cancer in both men and women [4]. Similarly, investigators in the Women’s Health Initiative Observational (WHIOB) Study found that higher HEI-2010, AHEI, aMED, and DASH scores were associated with lower risk of morality from all-cause and CVD but not for cancer [5].

It is unknown whether this association can be extrapolated to highly educated, low-risk populations such as a cohort of male physicians. Hence, our primary objective was to investigate the associations of the DASH, MED, and AHEI scores with mortality from all causes in a large prospective cohort of US male physicians. Our secondary objective was to investigate the association of these dietary patterns with CVD and cancer mortality.

## 2. Methods

### 2.1. Study Population

The Physicians’ Health Study (PHS) I and II are completed, randomized, double-blind, placebo-controlled trials designed to study low-dose aspirin, β-carotene, and vitamins for the primary prevention of CVD and cancer among US male physicians. Detailed descriptions of the PHS I and II have previously been published [6,7,8]. Of the 29,071 total participants in the PHS, 21,075 completed a food frequency questionnaire (FFQ) between 1999 and 2002 and were alive at baseline. We excluded individuals with missing data for any component of the MED, DASH, or AHEI indices (*n* = 5307); this resulted in a final sample of 15,768 for the current analyses. Each participant provided written, informed consent, and the institution review board at Brigham and Women’s Hospital approved the study protocol (2005P002385, 21 January 2021).

### 2.2. Ascertainment of Death in the PHS

Incidence of death was determined through annual follow-up questionnaires. A questionnaire was mailed to participants annually to collect data on new medical diagnoses, including death. Participants who did not return the questionnaires within 5–6 weeks were sent a follow-up questionnaire, for up to a maximum of four times to non-respondents. Subjects were called if they still did not answer the questionnaire. An endpoints’ committee confirmed death after review of the medical records. The deaths were categorized by type as due to cardiovascular disease, coronary artery disease, stroke, or cancer, based on review of autopsy reports, death certificates, medical records, or family/next-of-kin report [6,8,9,10]. Details on endpoint validation in the PHS have been published [8,9,10].

### 2.3. Assessment of Dietary Intake and Diet Scores

Diet information was obtained using a self-reported food frequency questionnaire (FFQ). For each individual food item, participants were asked to report their average consumption during the past year. Possible answers were never or less than once per month, 1–3/month, 1/week, 2–4/week, 5–6/week, 1/day, 2–3/day, 4–5/day, and 6+/day. The validity and reproducibility of FFQs have been published elsewhere [11,12].

### 2.4. MED Score 

We used a MED score, which was adapted for use in an American population by Fung et al. [13], based on the intake of nine items: vegetables, fruits, nuts, whole grains, legumes, fish, red meat, alcohol, and ratio of monounsaturated to saturated fat. The median intakes of MED components were as follows: vegetables, 2.20 servings per day; fruit, 2.13 servings per day; nuts, 0.067 serving per day; whole grains, 1.29 servings per day; legumes, 0.27 serving per day; fish, 0.14 serving per day; red meat, 0.56 serving per day; and monounsaturated to saturated fat ratio, 0.46. Participants with intake below the median intake received 0 point; otherwise, 1 point. One point was assigned for alcohol intake between 5 and 15 g/d. Hence, the score of MED diet ranged from 0 to 9.

### 2.5. DASH Score 

We used a version of the DASH score most commonly employed in studies of the U.S. population, as specified by Fung et al. [14]. DASH score includes eight components that are specified in DASH diet: fruits, vegetables, nuts + legumes, dairy, whole grains, red meat, sugar-sweetened beverage, and sodium intake. Scoring was done based on quintiles, with the lowest intake (quintile 1) receiving 1 point and quintiles with highest intake (quintile 5) receiving 5 points. For red meat, sugar-sweetened beverage, and sodium intake, scoring was done in reverse order, i.e., quintile 1 receiving 5 points and quintile 5 receiving 1 point. The DASH diet score ranged from 8 to 40.

### 2.6. AHEI Score 

The AHEI score was based on the intake of nine individual components, as earlier described by McCullough et al. [2]. The components include vegetables, fruits, nuts and soy, the ratio of white to red meat, cereal fiber, trans fat, the ratio of polyunsaturated fatty acids to saturated fatty acids, multivitamin use, and alcohol intake. Each component had the potential to contribute 0–10 points: A score of 0 represents the least healthy behavior, whereas a score of 10 indicates that the recommendations were fully met. Intermediate intakes were scored proportionately to receive points between 0 and 10. Multivitamin use was dichotomous, either 2.5 points (did not use) or 7.5 points (did use). All individual component scores were then summed to obtain an AHEI score, which ranged from 14.5 to 87.5.

### 2.7. Other Variables

Information on demographic variables, body mass index (BMI), cigarette smoking, exercise, and history of diabetes, hypertension, and atrial fibrillation were collected at baseline by self-reported questionnaire. All cardiovascular events including coronary heart disease (CHD) and congestive heart failure (CHF) have been adjudicated by an endpoint committee in the PHS [6,15]. Any participant reporting at least 1–3 drinks per month was classified as current drinker. At baseline, each subject was asked the following question: “How often do you exercise vigorously enough to work up sweat?” Possible answers included rarely/never, 1–3/month, 1/week, 2–4/week, 5–6/week, and daily]. Current exercise referred to people who reported at least 1–3 per month of vigorous exercise.

### 2.8. Statistical Analysis

Descriptive characteristics were used to estimate correlation coefficient between dietary indices. We created quintiles of each diet score and used Cox proportional hazard regression to estimate the hazard ratio of all-cause, CVD, and cancer mortality. We used two models: minimally adjusted model 1 (adjusted for age, energy, smoking, and exercise) and fully adjusted model 2 (adjusted for age, energy, smoking, exercise, BMI, and history of hypertension, atrial fibrillation, CHF, diabetes, and CHD). All analyses were completed using SAS, version 9.3 (SAS institute Inc, Cary, NC, USA). All *p*–values were two-tailed and significance level was set at an alpha of 0.05.

## 3. Results

Table 1 shows the baseline demographics of the 15,768 subjects according to quintile of diet scores. Mean age at FFQ was 65.9 ± 8.9 years (range: 50.2–97.6 years). During an average follow-up of 9.8 years, 1763 deaths were documented, including 488 CVD deaths and 589 cancer deaths. Across all dietary patterns, higher quintile of diet score was associated with older age, lower BMI, less-frequent smoking, more-frequent alcohol consumption, exercise, and higher energy intake. Subjects with higher quintile diet score were more likely to have CHD and atrial fibrillation and less likely to have diabetes. Subjects with higher AHEI and DASH quintile score were more likely to have hypertension and CHF, respectively. Spearman correlation coefficients were 0.71 between MED and DASH score (*p* < 0.001), 0.72 between MED and AHEI score (*p* < 0.001), and 0.66 between DASH and AHEI score (*p* < 0.001).

### 3.1. All-Cause Mortality

MED, DASH, and AHEI scores were inversely associated with risk for all-cause mortality (Table 2). Compared to quintile 1, MED scores in quintile 2, quintile 3, quintile 4, and quintile 5 were associated with 15% (HR 0.85; 95% CI 2–27%), 20% (HR 0.80; 95% CI 7–31%), 23% (HR 0.77; 95% CI 11–34%), and 32% (HR 0.68; 95% CI 21–42%) lower risk for all-cause mortality, respectively, (Table 2, *p* for trend < 0.001) in a model controlling for age, energy, smoking, exercise, BMI, and history of hypertension, diabetes, atrial fibrillation, CHF, and CHD. Similarly, in this multivariable model, DASH and AHEI scores were inversely associated with total mortality (Table 2, *p* for linear trend 0.019 and <0.001, respectively). 

### 3.2. CVD and Cancer Mortality

MED and AHEI scores were inversely associated with lower risk for CVD mortality (Table 3, *p* for linear trend 0.004 and <0.001, respectively). Similarly, AHEI score was inversely associated with lower risk for cancer mortality (Table 3, *p* for linear trend 0.008). MED and DASH scores were not associated with lower risk for cancer mortality.

## 4. Discussion

We found inverse associations of MED, DASH, and AHEI scores with total mortality in this population of US male physicians.

To the best of our knowledge, this is the first study to examine whether DASH, MED, and AHEI diet are associated with a lower risk of mortality in a low-risk, highly educated population of US male physicians. Our study was consistent with the previous research showing an inverse relationship between higher-quality dietary patterns and all-cause mortality [3,4,5], with some exceptions. In our study, the magnitude of all-cause mortality with AHEI diet score was larger compared to others (44% lower risk in our study compared to 24% in NIH-AARP, 22% in MEC, and 18% in WHIOB study). Similarly, the magnitude of all-cause mortality associated with higher MED score was larger in our study (32% compared to 23% in NIH-AARP, 22–24% in MEC, and 26% in WHIOB study). The magnitude of all-cause-associated lower risk of mortality with DASH was comparable to other studies. This could be explained by: (1) PHS is a highly educated cohort who most likely is on optimal medical treatment and more likely to follow dietary recommendations over a long period of time than subjects in other populations; hence, the mortality rate in the lowest quintile is more likely to be lower than in other cohorts; and (2) FFQ was collected between 1999–2002, whereas in other studies the FFQ was collected in early 1990s, hence, observing lower mortality rates across all quintiles with more awareness to have a healthy lifestyle and adhering to dietary recommendations.

Similarly, the magnitude of effect for CVD mortality with AHEI diet and MED diet was higher in our study compared to others. This might be due to a lower background rate in our cohort of male physicians. Higher DASH diet scores trended toward lower risk of CVD mortality in our study, but did not reach statistical significance, even in the model that did not adjust for hypertension, CHD, or congestive heart failure. A key feature of the DASH diet is its emphasis on low sodium intake, and randomized control trials have demonstrated that the DASH can produce significant reductions in blood pressure, even across a range of sodium intake [16,17]. Similar to other studies [18,19], the FFQ used in our study was not specifically designed to capture sodium intake and, thus, information regarding added sodium before or after prepared meals or the use of low-sodium foods was missing. We did not observe an association of DASH diet on cancer mortality. This is in line with the findings observed by Chiuve et al. [20], that cancer is a heterogeneous endpoint and different diets plays different roles in the etiology of certain cancers.

There are various biologic mechanisms by which healthy diet may lower the risk of mortality. Fruits and vegetables, whole grains, and unsaturated fatty acids may positively affect inflammation, vascular function, glucose-insulin homeostasis, and weight [21]. Nut consumption has been shown to lower blood pressure, improve lipid profiles [22,23], decrease inflammation [24], and increase insulin sensitivity [25]. Moderate alcohol consumption has been associated with lower risk of type 2 diabetes [26] and CVD [27,28]. These individual food items are key elements in MED, DASH, and AHEI diet patterns. Adherences to the MED and DASH diets are associated with decreased risk of cancer [29] and type 2 diabetes [30] and have demonstrated beneficial effects in prevention and treatment of CVD [14,31,32]. The AHEI diet has also been inversely associated with several chronic diseases, including diabetes mellitus [33] and CVD [34].

### Limitations

We did not have urine samples to quantify sodium intake in the PHS. Since the Physicians’ Health Study enrolled mostly Caucasian male physicians, the results from the study may not be generalizable to other populations, and the observational nature of our study prevents us from establishing causality between diet scores and mortality. Participants could have changed their dietary habits over time, leading to misclassification of dietary patterns and resulting in conservative estimate of the true effects of dietary patterns on mortality. Our study included subjects of well-nourished and highly educated physicians; these results cannot be generalized to other, poorly nourished, uneducated populations. Despite the above limitations, our study had numerous strengths including a large sample size, a lengthy follow-up period, standardized methods for data collection, comprehensive assessment of dietary habits, and robustness of the findings in multivariate analyses.

## 5. Conclusions

Overall, MED, DASH, and AHEI were each inversely associated with mortality from all causes in male physicians.

## Figures and Tables

**Table 1 nutrients-13-01893-t001:** Characteristics of the 15,768 study participants in the Physicians’ Health Study according to quintiles of the MED, DASH, and AHEI scores.

		Study Characteristics										
Diet (Quintiles)	Range of Diet Score	Age (years)	BMI (kg/m^2^)	Current Smoker (%)	Current Drinker (%)	Exercise (%)	Energy Intake (cal)	HTN (%)	DM (%)	AF (%)	CHF (%)	CHD (%)
MED												
1	0–2	64.9 ± 9.3	26.5 ± 3.7	5.2	77.9	52.3	1461 ± 419	45.4	7.7	7.7	1.9	10.7
2	3	65.8 ± 9.2	26.1 ± 3.4	3.9	79.9	59.2	1587 ± 464	46.0	7.6	8.6	1.5	11.1
3	4	65.9 ± 8.9	25.8 ± 3.3	3.4	82.6	62.9	1667 ± 496	46.4	7.1	8.3	2.0	12.3
4	5	66.5 ± 8.7	25.6 ± 3.3	2.5	83.4	66.8	1776 ± 513	45.3	7.0	10.0	2.0	13.6
5	6–9	66.6 ± 8.4	25.2 ± 3.0	1.6	84.3	70.7	1949 ± 540	46.6	6.5	10.1	1.9	15.3
DASH												
1	8–19	63.4 ± 8.5	26.8 ± 3.6	5.4	81.9	52.4	1451 ± 435	45.3	7.9	7.1	1.9	10.3
2	20–22	65.3 ± 8.8	26.3 ± 3.5	4.0	82.2	57.0	1576 ± 459	46.8	7.9	8.1	1.6	12.1
3	23–25	66.1 ± 8.9	25.9 ± 3.2	2.9	83.2	63.4	1678 ± 488	46.7	7.5	9.3	1.4	12.2
4	26–28	67.0 ± 8.9	25.4 ± 3.2	2.6	81.0	66.8	1810 ± 521	45.9	6.4	9.6	2.0	13.1
5	29–40	68.0 ± 8.9	24.8 ± 3.0	1.7	79.7	73.0	1972 ± 535	45.0	6.0	10.8	2.5	16.1
AHEI												
1	12.5–35.5	64.8 ± 9.2	26.5 ± 3.6	5.5	79.3	50.6	1434 ± 414	45.1	7.5	8.0	1.7	9.4
2	36.5–42.5	65.6 ± 9.2	26.2 ± 3.3	3.9	81.2	59.8	1618 ± 464	44.5	7.0	8.2	1.7	11.4
3	43.5–49.5	66.2 ± 9.0	25.9 ± 3.3	3.1	81.7	62.5	1721 ± 488	47.7	8.3	9.4	2.1	11.8
4	50.5–56.5	66.5 ± 8.7	25.6 ± 3.3	2.6	84.2	66.3	1804 ± 526	46.2	6.5	9.3	2.0	13.5
5	57.5–83.5	66.7 ± 8.3	25.0 ± 3.1	1.4	82.0	73.4	1903 ± 571	46.1	6.4	9.9	1.9	17.6

Data are mean ± SD for continuous variables or % for categorical variables. Abbreviations: AF, atrial fibrillation; AHEI, Alternative Healthy Eating Index; BMI, body mass index; CHD, coronary heart disease; CHF, congestive heart failure; DASH, Dietary Approach to Stop Hypertension; DM, diabetes mellitus; HTN, hypertension; MED, Mediterranean.

**Table 2 nutrients-13-01893-t002:** HR (95% CIs) for all-cause mortality according to quintiles of MED, DASH, and AHEI scores among 15,768 men in the Physicians’ Health Study.

Quintiles	#Cases/*n*	Model 1 ^a^	Model 2 ^b^
MED diet score			
1	444/3467	1.00	1.00
2	324/2751	0.83 (0.72–0.96)	0.85 (0.73–0.98)
3	336/2969	0.81 (0.71–0.94)	0.80 (0.69–0.93)
4	285/2611	0.78 (0.67–0.91)	0.77 (0.66–0.90)
5	374/3970	0.68 (0.59–0.79)	0.68 (0.58–0.79)
*p* for trend		<0.001	<0.001
DASH diet score			
1	300/3069	1.00	1.00
2	342/3068	0.95 (0.81–1.11)	0.96 (0.82–1.12)
3	411/3514	0.95 (0.82–1.10)	0.95 (0.82–1.11)
4	353/3050	0.87 (0.74–1.02)	0.88 (0.75–1.04)
5	357/3067	0.83 (0.71–0.98)	0.83 (0.71–0.99)
*p* for trend		0.018	0.019
AHEI diet score			
1	422/3089	1.00	1.00
2	403/3111	0.89 (0.77–1.02)	0.88 (0.77–1.02)
3	411/3434	0.84 (0.73–0.97)	0.82 (0.71–0.95)
4	286/2954	0.72 (0.61–0.84)	0.69 (0.59–0.81)
5	241/3180	0.58 (0.49–0.69)	0.56 (0.47–0.67)
*p* for trend		<0.001	<0.001

^a^ Hazard ratio (HR) with 95% confidence interval (CI) from a model including age, energy, smoking, and exercise. ^b^ Hazard ratio (HR) with 95% confidence interval (CI) from a model including for age, energy, smoking, exercise, body mass index, and history of hypertension, atrial fibrillation, congestive heart failure, diabetes, and coronary heart disease. Abbreviations: AHEI, Alternative Healthy Eating Index; DASH, Dietary Approach to Stop Hypertension; MED, Mediterranean.

**Table 3 nutrients-13-01893-t003:** HR (95% CIs) for CVD and cancer mortality according to quintiles of MED, DASH, and AHEI scores among 15,768 men in the Physicians’ Health Study.

Quintiles	#Cases/*n*	CVD Mortality	#Cases/*n*		Cancer Mortality	
		Model 1 ^a^	Model 2 ^b^		Model 1 ^a^	Model 2 ^b^
MED diet score						
1	113/3467	1.00	1.00	143/3467	1.00	1.00
2	98/2751	0.97 (0.74–1.27)	0.99 (0.75–1.30)	112/2751	0.91 (0.71–1.16)	0.92 (0.72–1.19)
3	101/2969	0.94 (0.71–1.23)	0.91 (0.69–1.20)	100/2969	0.77 (0.60–1.00)	0.79 (0.61–1.03)
4	72/2611	0.75 (0.56–1.02)	0.73 (0.54–0.99)	111/2611	0.96 (0.74–1.24)	1.0 (0.77–1.29)
5	104/3970	0.72 (0.54–0.95)	0.71 (0.53–0.94)	123/3970	0.72 (0.56–0.93)	0.76 (0.59–0.99)
*p* for trend		0.007	0.004		0.037	0.104
DASH diet score						
1	77/3069	1.00	1.00	103/3069	1.00	1.00
2	105/3068	1.08 (0.80–1.46)	1.14 (0.85–1.54)	108/3068	0.92 (0.70–1.20)	0.93 (0.70–1.22)
3	109/3514	0.94 (0.70–1.27)	0.98 (0.73–1.32)	144/3514	1.02 (0.79–1.32)	1.06 (0.82–1.37)
4	98/3050	0.88 (0.64–1.20)	0.91 (0.67–1.25)	119/3050	0.93 (0.71–1.23)	0.99 (0.75–1.30)
5	99/3067	0.82 (0.60–1.13)	0.85 (0.62–1.17)	115/3067	0.88 (0.66–1.17)	0.94 (0.70–1.25)
*p* for trend		0.088	0.120		0.430	0.800
AHEI diet score						
1	112/3089	1.00	1.00	133/3089	1.00	1.00
2	105/3111	0.85 (0.65–1.12)	0.84 (0.64–1.10)	147/3111	1.06 (0.84–1.35)	1.08 (0.85–1.38)
3	123/3434	0.93 (0.72–1.21)	0.88 (0.67–1.14)	120/3434	0.81 (0.63–1.04)	0.84 (0.65–1.08)
4	81/2954	0.74 (0.55–1.00)	0.71 (0.52–0.95)	102/2954	0.83 (0.63–1.09)	0.86 (0.66–1.13)
5	67/3180	0.59 (0.43–0.81)	0.55 (0.40–0.76)	87/3180	0.68 (0.51–0.91)	0.73 (0.54–0.97)
*p* for trend		0.001	<0.001		0.002	0.008

^a^ Hazard ratio (HR) with 95% confidence interval (CI) from a model including for age, energy, smoking, and exercise. ^b^ Hazard ratio (HR) with 95% confidence interval (CI) from a model including for age, energy, smoking, exercise, body mass index, and history of hypertension, atrial fibrillation, congestive heart failure, diabetes, and coronary heart disease. Abbreviations: AHEI, Alternative Healthy Eating Index; CVD, cardiovascular disease; DASH, Dietary Approach to Stop Hypertension; MED, Mediterranean.

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
