# Peer review of "Mediterranean, DASH, and Alternate Healthy Eating Index Dietary Patterns and Risk of Death in the Physicians’ Health Study"

_nutrients, 2021, doi:10.3390/nu13061893_

Round 1

Reviewer 1 Report

I am quite surprised by the authors' decision not to include racial/ethnic demographics in the analysis of this data. This is a correctable error and it would contribute a great deal to this paper. These diets have already been compared, and the study doesn't add a significant new findings. In fact, the findings are somewhat skewed because physicians enjoy an income that is able to support optimal dietary choices, they have ample access to healthcare, and as an all male sample, their heart disease is taken more seriously sooner. These limitations are not included in the discussion.

Author Response

We want to thank this reviewer for their comments and review.

Comment 1. I am quite surprised by the authors' decision not to include racial/ethnic demographics in the analysis of this data. This is a correctable error and it would contribute a great deal to this paper. These diets have already been compared, and the study doesn't add a significant new findings. In fact, the findings are somewhat skewed because physicians enjoy an income that is able to support optimal dietary choices, they have ample access to healthcare, and as an all male sample, their heart disease is taken more seriously sooner. These limitations are not included in the discussion.

Answer. We have acknowledged in limitation section in lines 243-245 that Physicians’ Health Study was a study done primarily in Caucasian (>90%)  males and hence results are not generalizable to the entire population. While having data in other racial/ethnic groups would have been very helpful and meaningful, we unfortunately do not have the data in other racial/ethnic groups to do these sensitivity analysis. Physicians’ Health Study was primarily done in male physicians and we do not have data on women physicians to do a sensitivity analysis. Similarly, given majority of Physicians enrolled into the study were Caucasians, we do not have enough data to do sensitivity analysis amongst other ethnicities. However, as described in lines 47-52, in Women’s Health Initiative Observational Study, higher diet scores were seen to be associated with lower risk of mortality from all-cause and cardiovascular disease amongst women. The result from current study shows similar results amongst males and hence such findings could be generalized to both genders. We still feel our manuscript provides important result findings amongst Physicians with higher education and higher income category, who are widely underrepresented in other study populations. We have added in our limitation in lines 249-251 that our results cannot be generalized to other poorly nourished and uneducated population as suggested by this reviewer.

Reviewer 2 Report

Overall the paper is interesting and well written, however some points need to be verified:

1. Methods lines 58-60: The Physicians’ Health Study (PHS) I and II are completed, randomized, double-58 blind, placebo-controlled trial designed to study low-dose aspirin, β-carotene, and vita-59 mins for the primary prevention of CVD and cancer among US male physicians.

There is nothing mentioned about the possible impact of supplementation on the study results.

2. Methods, please add criteria how were classified individuals to the current drinkers and individuals to the exercise group.

3. Results line 137 “… with higher AHEI and DASH quintile score were more likely to have hypertension…”. According to the table 1, the HTN prevalence across quintiles of DASH scores was 45.3; 46.8; 46.7; 45.9; 45.0 respectively. I suggest to add statistical analysis to table 1.

4. Results line 150 “and quintile 5 were associated with 16% (HR 0.85; 95%CI 2%-27%),…” why 16% not 15%? (HR is 0.85)

5. Results lines 164-165 “MED, DASH, and AHEI scores were inversely associated with lower risk for CVD 164 mortality (Table 3, p for linear trend 0.004, 0.12, and 0.0002 respectively).” The results for DASH scores were not statistically significant (p=0.12).

6. Abstract line 29 – the same comment as in point 5. DASH scores were not significantly associated with risk for CVD mortality.

7. There are some typos in the text, for example: table 2 “mode 1” (it should be model) or table 3 “ #Case/n” (it should be Cases).

Author Response

We want to thank this reviewer for their insightful comments and review.

Overall the paper is interesting and well written, however some points need to be verified

Answer. We want to thank you for this reviewer’s complements.

1. Methods lines 60-62: The Physicians’ Health Study (PHS) I and II are completed, randomized, double- blind, placebo-controlled trial designed to study low-dose aspirin, β-carotene, and vitamins for the primary prevention of CVD and cancer among US male physicians.

There is nothing mentioned about the possible impact of supplementation on the study results.

Answer: We do not think there is possibility of any impact of low dose aspirin, β-carotene, or any other vitamins on our study results. Given the primary analyses in Physicians’ Health Study I and II were negative on impact of aspirin, β-carotene, and vitamins to total mortality or CVD mortality, we do not think there would be any impact of low dose aspirin, β-carotene, or any other vitamins on our study results.

2. Methods, please add criteria how were classified individuals to the current drinkers and individuals to the exercise group.

Answer. Any participant  reporting at least 1-3 drinks per month was classified as current drinker.

At baseline, each subject was asked the following question: “How often do you exercise vigorously enough to work up sweat?” Possible answers included rarely/never, 1–3/month, 1/week, 2–4/week, 5–6/week, and daily]. Current exercise referred to people that reported at least 1-3 per month of vigorous exercise.

We have added these information in lines 120-123.

3. Results line 137 “… with higher AHEI and DASH quintile score were more likely to have hypertension…”. According to the table 1, the HTN prevalence across quintiles of DASH scores was 45.3; 46.8; 46.7; 45.9; 45.0 respectively. I suggest to add statistical analysis to table 1.

Answer: It seems that the reviewer might have misinterpreted this sentence in manuscript. As written in lines 141-143, we have mentioned that subjects with higher AHEI diet scores are more likely to have hypertension and subjects with higher DASH scores are more likely to have CHF. We did not say that the subjects with higher DASH scores are more likely to have HTN as the prevalence of HTN is comparable across quintiles. We do not think adding statistical analysis for table 1 will add any more information than what have already been provided and since p value no matter how small does not say anything about confounding  in the data.

4. Results line 150 “and quintile 5 were associated with 16% (HR 0.85; 95%CI 2%-27%),…” why 16% not 15%? (HR is 0.85)

Answer. We want to thank you to this reviewer for diligent observation. This was an oversight on our part. We have corrected this to be 15% in our manuscript.

5. Results lines 164-165 “MED, DASH, and AHEI scores were inversely associated with lower risk for CVD mortality (Table 3, p for linear trend 0.004, 0.12, and 0.0002 respectively).” The results for DASH scores were not statistically significant (p=0.12).

Answer: We have revised the sentence to exclude non-statistically significant trend for DASH in lines 175-176.

6. Abstract line 29 – the same comment as in point 5. DASH scores were not significantly associated with risk for CVD mortality.

Answer: We have revised this in the abstract in line 28-29.

7. There are some typos in the text, for example: table 2 “mode 1” (it should be model) or table 3 “ #Case/n” (it should be Cases).

Answer. Thank you. We have corrected these typos.

Reviewer 3 Report

This is an interesting study, examining the impact of three different dietary patterns on mortality.

In order to provide an addition to the field, it would be interesting to examine which dietary pattern best predict mortality outcomes, e.g. examining differences in the odds in the 5th quintile among dietary patterns. 

In addition to existing tables, survival plots could offer a better presentation of the results. 

Minor comments:

Abstract, lines 22-6: it would be more clear to the reader if you provide only values for the 5th quintile

Results, lines 133-7: where these difference statistically significant?

Table 1, column "Exercise": physical activity assessment is not presented in the methods section

All tables and text: provide all p-values with three decimal places (or at least with the same throughout)

Author Response

We want to thank this reviewer for their insightful comments and review.

This is an interesting study, examining the impact of three different dietary patterns on mortality.

Answer. We want to thank you to this reviewer for encouraging our efforts to publish this data.

In order to provide an addition to the field, it would be interesting to examine which dietary pattern best predict mortality outcomes, e.g. examining differences in the odds in the 5th quintile among dietary patterns. 

Answer. This is an interesting question raised by the reviewer. As shown in Table 2, improvement amongst all three diet patterns is associated with lower risk of mortality. We do not feel that one dietary pattern is better than other. Whichever diet pattern you chose, improvement in that dietary pattern is associated with lower mortality.

In addition to existing tables, survival plots could offer a better presentation of the results. 

Answer. Thank you for this comment. We feel that describing our results will be redundant if we are to show Table 2 and Table 3 with Hazard Ratios and survival plot. Hence, we did not show results in form of a survival plot.

Abstract, lines 22-6: it would be more clear to the reader if you provide only values for the 5th quintile

Answer. As explained by us earlier, we want to emphasize  that improvement in diet scores of any diet pattern is associated with lower risk of mortality. Even if you improve your dietary patten from Quintile 1 to Quintile 2, your effort in dietary improvement is not wasted and that it helps to reduce the risk of mortality.

Results, lines 133-7: where these difference statistically significant?

Answer. Table 1 shows demographics of our study population at baseline. We strongly feel it is meaningless to show statistical analysis showing correlation between diet scores and baseline variables. Hence, intentionally we have not included P-values in Table 1 as p value does not have bearing on confounding in the data. For example, p value of 0.8 for age difference does not exclude confounding by age.

Table 1, column "Exercise": physical activity assessment is not presented in the methods section

Answer. At baseline, each subject was asked the following question: “How often do you exercise vigorously enough to work up sweat?” Possible answers included rarely/never, 1–3/month, 1/week, 2–4/week, 5–6/week, and daily]. Current exercise referred to people that reported at least 1-3 per month of vigorous exercise.

We have added these information in lines 121-123.

All tables and text: provide all p-values with three decimal places (or at least with the same throughout)

Answer. We have rectified to have three decimal places throughout the manuscript in order to be consistent in our reporting P values.

Round 2

Reviewer 1 Report

no additional observations